# MPE-HRNet^*L*^: A Lightweight High-Resolution Network for Multispecies Animal Pose Estimation

**DOI:** 10.3390/s24216882

**Published:** 2024-10-26

**Authors:** Jiquan Shen, Yaning Jiang, Junwei Luo, Wei Wang

**Affiliations:** 1School of Software, Henan Polytechnic University, Jiaozuo 454000, China; sjq@hpu.edu.cn (J.S.); luojunwei@hpu.edu.cn (J.L.); 2Anyang Institute of Technology, Anyang 455000, China; 3School of Computer Science and Technology, Henan Polytechnic University, Jiaozuo 454000, China; 212209010005@home.hpu.edu.cn; 4CSSC Haifeng Aviation Technology Co., Ltd., No.4, Xinghuo Road, Fengtai Science Park, Beijing 100070, China

**Keywords:** animal pose estimation, multiscale mixed attention mechanism, high-resolution network, spatial pyramid pooling, Lite-HRNet

## Abstract

Animal pose estimation is crucial for animal health assessment, species protection, and behavior analysis. It is an inevitable and unstoppable trend to apply deep learning to animal pose estimation. In many practical application scenarios, pose estimation models must be deployed on edge devices with limited resource. Therefore, it is essential to strike a balance between model complexity and accuracy. To address this issue, we propose a lightweight network model, i.e., MPE-HRNet.L, by improving Lite-HRNet. The improvements are threefold. Firstly, we improve Spatial Pyramid Pooling-Fast and apply it and the improved version to different branches. Secondly, we construct a feature extraction module based on a mixed pooling module and a dual spatial and channel attention mechanism, and take the feature extraction module as the basic module of MPE-HRNet.L. Thirdly, we introduce a feature enhancement stage to enhance important features. The experimental results on the AP-10K dataset and the Animal Pose dataset verify the effectiveness and efficiency of MPE-HRNet.L.

## 1. Introduction

Animal pose estimation aims to identify and locate the key points of animal bodies. It plays a crucial role in assessing animal health [1], protecting endangered species [2], and analyzing animal behavior [3]. With the advancement of deep learning technology, it has been an inevitable trend to apply the technology to animal pose estimation. Deep-learning-based solutions can be categorized into bottom-up solutions and top-down solutions [4]. The bottom-up solutions detect key points and then cluster the key points to generate specific poses [5,6]. Although these solutions are relatively fast, they cannot guarantee accuracy in complex scenes. Different from the bottom-up solutions, the top-down solutions [7,8,9] identify individual animal objects, and then estimate the poses of these objects, yielding higher accuracy [5,6].

A lot of effort has been devoted to the top-down solutions [10,11]. For example, some researchers utilize large network models, such as Hourglass [12] and HRNet [8], to achieve higher accuracy at the cost of high complexity [13,14,15,16,17,18,19]. In many practical application scenarios, pose estimation models must be deployed on edge devices. Edge devices are always equipped with limited resource. They are unable to leverage the advantages of large models since they cannot provide the resource that large models need. Inspired by ShuffleNet [20], Yu et al. [9] proposed a lightweight version of HRNet, i.e., Lite-HRNet. Lite-HRNet can estimate the poses of human beings with relatively high speed and accuracy. However, it shows an obvious drop in accuracy when estimating the poses of animals of multiple species. That happens because of two reasons. The first reason is that animals of different species possess different anatomical structures, resulting in significant variations in their poses. The second reason is that Lite-HRNet sacrifices accuracy in favor of a light network structure.

In this paper, we propose a lightweight network to estimate the poses of multiple species on edge devices with higher accuracy. We name the lightweight network MPE-HRNet.L and construct it by improving Lite-HRNet from three aspects. Firstly, we improve Spatial Pyramid Pooling-Fast (SPPF) and propose SPPF.+. We apply SPPF and SPPF.+ to different branches in order to enrich the receptive fields of feature maps and mitigate the impact of feature loss caused by pooling operations. Secondly, we propose a mixed pooling module (MPM) and a dual spatial and channel attention mechanism (DSCA). Leveraging MPM and DSAM, we construct a mixed feature extraction (MFE) block with two different versions. Thirdly, we design a feature enhancement (FE) stage, and append it as the last stage of MPE-HRNet.L. The main contributions of this work are as follows.
We propose SPPF.+. Comparing with SPPF, SPPF.+ has a stronger feature extraction capability and lower computational load.We propose MPM and DSCA. Based on MPM and DSCA, we design MFE block which possesses strong feature extraction capabilities.We design the FE stage and take it as the final stage of MPE-HRNet.L. This stage is introduced to emphasize the semantic feature in the output feature map of MPE-HRNet.L.

The rest of this paper is organized as follows. Section 2 introduces the theoretical basis. Section 3 elaborates MPE-HRNet.L. After the experimental results are presented in Section 4, the paper is concluded in Section 5.

## 2. Related Works

This section presents the significant advancements in animal pose estimation and two important techniques of deep learning. Firstly, Section 2.1 summarizes the research status of animal pose estimation. Secondly, Section 2.2 addresses the evolution and application of attention mechanisms. Finally, Section 2.3 discusses the advantage and the development of spatial pyramid pooling (SPP).

### 2.1. Animal Pose Estimation

Traditional solutions [21] for animal pose estimation involve obtaining behavioral data through manual observation or attaching sensors to animals. These methods incur high labor costs and implementation difficulties, along with challenges related to equipment vulnerability and data collection for rare species. Deep-learning-based pose estimation solutions can effectively alleviate these problems [22]. These solutions can be classified into bottom-up solutions and top-down solutions. Considering accuracy, this paper solely focuses on the top-down solutions.

To improve accuracy, large network models have been introduced to some of the top-down solutions. Graving et al. [17] developed a software toolkit based on Hourglass and DenseNet [23] to estimate keypoint locations with subpixel precision. Zhou et al. [16] improved Hourglasses to model the differences and spatial correlations between mouse body parts. Wang et al. [13] designed a grouped pig pose estimation model which exploited HRNet [24] to predict the heatmap of keypoints of pigs. Fan et al. [14] and Zhao et al. [18] also exploited HRNet. They constructed a pose estimation network for cattle and Jinling ducks, respectively. He et al. [15] combined Vision Transformer (ViT) [25] and HRNet, and constructed a model for birds. The above solutions are only applicable in scenarios involving single-species pose estimation.

In multispecies pose estimation scenarios, Gong et al. [19] applied dynamic convolution and a receptive field block to HRNet, thereby significantly enhancing the feature extraction of quadrupeds. In order to address the challenges in animal data annotation, Liao et al. [26] proposed THANet to transfer human pose estimation to animal pose estimation. Different from the solution of Liao et al. [26], Hu and Liu [27] utilized a language–image contrastive learning model to address the challenges, while Zeng et al. [28] employed a semi-supervised method with MVCRNet to utilize unlabeled data. All these solutions exploit large models. In many practical application scenarios, models are always deployed on edge devices. Because of the limited memory and computing power, large models are not suitable for those devices. We try to make a trade-off between accuracy and complexity, and design a lightweight model to run animal pose estimation on edge devices with relatively high accuracy.

### 2.2. Attention Mechanism

Attention mechanism [29] allows models to focus on important areas in an image by mimicking the human visual system. It calculates a weight matrix for an input feature map and depends on the weight matrix to highlight the important features and weaken the irrelevant features in the feature map. SENet [30], ECANet [31], FcaNet [32], and CBAM [33] are all attention mechanisms.

SENet is composed of a squeeze module and an excitation module. The squeeze module extracts global spatial information from input feature maps and then compresses these feature maps using global average pooling. The excitation module uses a fully connected layer to capture the relationships among these compressed feature maps and output an attention weight matrix for the input feature maps. Wang et al. [31] believe that the fully connected layer in the excitation module increases the complexity of SENet. They replaced the fully connected layer with a 1D convolution, from which they came up with ECANet. Qin et al. [32] also made improvements to SENet. They replaced the global average pooling with the following operations: grouping the feature maps and then processing the grouped feature maps with a 2D discrete cosine transform function. CBAM is a hybrid attention mechanism that combines the channel features and spatial features of input feature maps to figure out where to focus and what to focus on.

### 2.3. Spatial Pyramid Pooling

Convolutional neural networks (CNNs) [34] often require input images to meet a special requirement of size to ensure that the fully connected layer can output feature vectors of a specific size. Traditional methods exploit cropping and stretching to generate images of a specific size, but this can result in image distortion and information loss. SPP [35] utilizes pooling operations of different scales to transform images of different sizes into a specific size, thus avoiding stretching or cropping images. Chen et al. [36] replaced the pooling layers in SPP with multiple dilated convolutions with different dilation factors to generate feature maps with larger receptive fields. Jocher [37] improved SPP and proposed SPPF. SPPF exploits multiple concatenated pooling layers of the same scale to process input feature maps, and fuses the pooling results in parallel to improve speed.

## 3. Proposed Algorithm

In this section, we illustrate the proposed MPE-HRNet.L. We outline the structure of MPE-HRNet.L in Section 3.1, and then elaborate the improved version of SPPF in Section 3.2. After representing the details of MFE block in Section 3.3, we describe the feature enhancement stage in Section 3.4.

### 3.1. Architecture of MPE-HRNet.L

To construct a lightweight network for accurately estimating the poses of multiple animal species, we improved Lite-HRNet and proposed MPE-HRNet.L. The improvements include optimizing branches with SPPF, SPPF.+ and MFE block, and introducing a feature enhancement (FE) stage. The branch optimization is designed to enhance the feature extraction and representation capabilities of MPE-HRNet.L. The introduction of the FE stage aims to enhance the important features in the final outputs of MPE-HRNet.L. Figure 1 and Table 1 show the architecture and the details of MPE-HRNet.L, respectively.

As Figure 1 and Table 1 show, stage stem only creates a new branch with a conv2d block and a shuffle block. Both stage 2 and stage 3 create a new branch, and apply SPPF.+ to the branches created by the previous stage of each of them. They stack multiple MFE blocks to extract features from all the branches except the ones created by themselves. Similar to stage 2 and stage 3, stage 4 also applies stacked MFE blocks to branches. The difference is that stage 4 does not create any new branches, and it applies SPPF instead of SPPF.+ to the branch created by its previous stage. Stage FE is quite different from the former four stages. It is introduced to enhance the features output by stage 4.

### 3.2. SPPF^+^

Lite-HRNet utilizes downsampling to create new branches. However, downsampling can only generate feature maps with limited receptive fields. It could probably also lose some important features. SPPF can generate feature maps with rich receptive fields by extracting multiple features of varying scales and fusing them together. However, SPPF has three obvious limitations. Firstly, 1 × 1 convolution struggles to extract complex features. Secondly, pooling operations also lose some features. Finally, concat operation increases computational complexity. To address these issues, we improve SPPF and propose SPPF.+.

In order to enhance the ability to extract features, SPPF.+ replaces the 1 × 1 convolution with a depthwise separable convolution. To ensure that the feature maps generated by pooling operations contain rich features, SPPF.+ inserts fusion operations among the sequential pooling operations. Concretely, it fuses the results of the initial two pooling operations, and then processes the fused results with the last pooling operation. To reduce the computational complexity caused by the concatenation operation, SPPF.+ refrains from concatenating the feature maps produced by the pooling operations with the one yielded by the depthwise separable convolution. Instead, it fuses the results of the last pooling operation with the fused results of the first two pooling operations, and then concatenates the fused results with the results of the depthwise separable convolution. SPPF.+ employs ADD operations to achieve fusion.

Figure 2 illustrates the architecture of SPPF.+. $sppf^{+}\left(X_{s}\right)$ denotes the outputs of SPPF.+ with the given input $X_{s}$, which includes multiple channels. It is computed according to Equation (Equation 1). In the equation, $conv_{1\times 1}\left(\cdot \right)$ represents 1 × 1 convolution, concat(·) denotes the concatenation operation, $conv_{DC}\left(\cdot \right)$ describes depthwise separable convolution, and $mmap_{p}\left(\cdot \right)$ represents a series of max pooling operations. $mmap_{p}\left(\cdot \right)$ is calculated according to Equations (Equation 2)–(Equation 5).
(1)$$sppf^{+}\left(X_{s}\right)=conv_{1\times 1}(concat\left(conv_{DC}\left(X_{s}\right),mmap_{p}\left(X_{s}\right)\right)$$
(2)$$mmax_{p}\left(X_{s}\right)=smax_{p}\left(X_{s}\right)+dmax_{p}\left(X_{s}\right)+tmax_{p}\left(X_{s}\right)$$
(3)$$smax_{p}\left(X_{s}\right)=max_{p}\left(conv_{DC}\left(X_{s}\right)\right)$$
(4)$$dmax_{p}\left(X_{s}\right)=max_{p}\left(smax_{p}\left(X_{s}\right)\right)$$
(5)$$tmax_{p}\left(X_{s}\right)=max_{p}\left(smax_{p}\left(X_{s}\right)+dmax_{p}\left(X_{s}\right)\right)$$

### 3.3. Mixed Feature Extraction Block

Lite-HRNet exploits the conditional channel weight block (CCW) to extract features. The CCW includes two weight calculation functions, $H_{s}$ and $F_{s}$. $H_{s}$ employs average pooling to generate low-resolution feature maps, which may loss some important details. $F_{s}$ neglects spatial information when computing weight matrices, which is crucial for localizing keypoints. To address these issues, we propose WGM along with DSCA, and construct the MFE block based on them, which serves as the foundational module for feature extraction.

#### 3.3.1. Weight Generation Module

To alleviate the feature loss caused by average pooling operation in $H_{s}$, we propose a mixed pooling module (MPM) and construct WGM by replacing the pooling operation in $H_{s}$ with MPM. Figure 3 shows the structure of WGM.

In the figure, the area surrounded by dashed lines illustrates the details of MPM. As the figure shows, MPM consists of a context-aware branch, an average pooled branch, and a maximum pooled branch. The three branches are designed to capture spatial contextual features, highlight local prominent features, and preserve global features, respectively.

The context-aware branch uses a point-wise convolution to extract spatial contextual features, followed by a softmax activation to generate a weight matrix. Subsequently, the weight matrix and the input feature maps undergo matrix transformations and multiplication to map the spatial contextual features into the input feature maps.

Taking $X_{m}$ as the input of MPM, $X_{m}$ can be described as ($X_{m_{0}},X_{m_{1}},\cdots ,X_{m_{c-1}}$), where *c* denotes the total number of channels in $X^{m}$. The output of MPM is denoted as $mpm(X_{m})$. It is calculated according to Equation (Equation 6). In the equation, $mbrch(X_{m},i)$ denotes the results of the i-th branch in MPM. It is calculated by Equation (Equation 7). In the equation, gap(·) and gmp(·) separately describe global average pooling and global max pooling; δ(·) denotes the softmax function, and ${\left(\cdot \right)}^{T}$ represents matrix transformation.
(6)$$mpm\left(X_{m}\right)=\sum_{i=0}^{2}mbrch\left(X_{m},i\right)\times \theta_{i}$$
(7)$$mbrch\left(X_{m},i\right)=\left\{\begin{array}{cc} {\left({\left(\delta \left({\left(conv_{1\times 1}\left(X_{m}\right)\right)}^{T}\right)\right)}^{T}\times X_{m}^{T}\right)}^{T}, & i=0\\ gap(X_{m}), & i=1\\ gmp(X_{m}), & i=2 \end{array}\right.$$

Figure 3 also shows the structure of WGM. According to the figure, WGM first performs mixed pooling on input feature maps and then concatenates the pooled results. After that, it processes the concatenated results using two sequential 1 × 1 convolutions, followed by a sigmoid function to the outputs of the convolutions to generate weight matrices.

Taking $X_{w}$ as the input of WGM, $X_{w}$ can be described as ($X_{w_{0}},X_{w_{1}},\cdots ,X_{w_{b-1}}$), where *b* denotes the total number of the existing branches in the corresponding stage. Moreover, $\forall X_{w_{i}}\left(X_{w_{i}}\in X_{w}\right)$, $X_{w_{i}}$ comprises multiple channel feature maps. For the given input $X_{w}$, the result of WGM is denoted as $wgm(X_{w})$. $wgm(X_{w})$ is calculated by Equation (Equation 8). In the equation, ups(·) denotes the upsampling operation, and σ(·) represents the sigmoid function. $\forall X_{w_{i}}\left(X_{w_{i}}\in X_{w}\right)$, WGM calculates a weight matrix for each channel feature map in $X_{w_{i}}$.
(8)$$wgm\left(X_{w}\right)=ups\left(\sigma \left(con_{1\times 1}\left(con_{1\times 1}\left(concat\left(X_{w}\right)\right)\right)\right)\right)$$

#### 3.3.2. Dual Spatial and Channel Attention Module

Each stage includes both low-resolution branches and high-resolution branches with channel and spatial features. Channel features are richer than spatial features in low-resolution feature maps, while both channel features and spatial features are rich in high-resolution feature maps. $F_{s}$ calculates weights based solely on channel features. It does not capitalize on spatial features which are vital for keypoint localization. To leverage both channel and spatial features for weight calculation, we introduce a dual spatial attention module (DSA) and a hybrid attention mechanism, i.e., DSCA. DSCA is constructed by combining DSA with a channel attention module. Figure 4 shows the structure of DSCA.

DSA utilizes four branches to extract spatial features. The first branch processes each input feature map with max and average pooling, then performs a weighted sum. The second, third, and fourth branches separately use a 1 × 1 convolution, a 3 × 3 convolution, and a 5 × 5 convolution to process each feature map and generate spatial features of various receptive fields. The first and second branches extract global spatial features, capturing the overall structure and layout relationships in each input feature map to obtain higher-level, more abstract semantic information. The third and fourth branches focus on extracting local spatial features, enhancing the attention to different regions within an input image.

After extracting spatial features, DSA concatenates the output of each branch and employs a 7 × 7 convolution to process the concatenated results. It generates a spatial attention weight matrix by applying sigmoid activation function to the results of the 7 × 7 convolution. Given the input $X_{d}$ with *c* channels, the result of DSA is calculated according to Equation (Equation 9). In the equation, σ denotes sigmoid function, $conv_{7\times 7}$ describes the 7×7 convolution, $concat_{i=0}^{3}\left(\cdot \right)$ represents the operation to concatenate the results of the four branches, and $dbrch(X_{d},i)$ depicts the results of the i-th branch. $dbrch(X_{d},i)$ is calculated according to Equation (Equation 10) in which mp(·) and ap(·) describe max pooling and average pooling, respectively.
(9)$$dsa\left(X_{d}\right)=\sigma \left(conv_{7\times 7}\left(concat_{i=0}^{3}\left(dbrch\left(X_{d},i\right)\right)\right)\right)$$
(10)$$dbrch\left(X_{d},i\right)=\left\{\begin{array}{cc} 0.5(mp\left(X_{d}\right)+ap\left(X_{d}\right)), & i=0\\ conv_{1\times 1}\left(X_{d}\right), & i=1\\ conv_{3\times 3}\left(X_{d}\right), & i=2\\ conv_{5\times 5}\left(X_{d}\right), & i=3 \end{array}\right.$$

dsca(·) denotes the outputs of DSCA. It can be calculated according to Equation (Equation 11). In this equation, ca(·) represents the output of a channel attention module, and ⊙ denotes element-wise multiplication. To simultaneously utilize both channel and spatial features for weight calculation, we utilize DSCA to replace $F_{s}$ to compute weights.
(11)$$dsca\left(X_{d}\right)=ca\left(X_{d}\right)⊙dsa\left(ca\left(X_{d}\right)⊙X_{d}\right)$$

#### 3.3.3. Mixed Feature Extraction Block

We construct MFE block by substituting $H_{s}$ with WGM and $F_{s}$ with DSCA, respectively. MFE has two architectures, namely, MFE-H and MFE-L, which utilize WGM and DSCA in different manners. MFE-H utilizes both WGM and DSCA, while MFE-L only uses WGM. Figure 5 compares the structures of MFE block with CCW block.

High-resolution branches are rich in both channel and spatial features, while low- resolution branches are rich only in channel features. Thus, we apply MFE-H to the two highest-resolution branches and MFE-L to the others.

### 3.4. Feature Enhancement Stage

Lite-HRNet uses bilinear interpolation in stage 4 to perform upsampling and feature fusion step by step from the 1/32 branch, until generating the output feature maps of the 1/4 branch, which is used for generating heatmaps and locating keypoints. However, bilinear interpolation only considers the subpixel neighborhood and generates feature maps within small receptive fields, which is incapable of capturing the rich semantic information needed for pose estimation. Therefore, we introduce the FE stage to capture more semantic information.

Figure 6 illustrates the structure of the FE stage. The FE stage utilizes content-aware reassembly of features (CARAFE) [38] to upsample input feature maps from 1/8 to 1/4 resolution. Then, it uses the efficient channel attention module (ECA) to process the generated feature maps, and fuses the processed results with the input feature maps at 1/4 resolution. Finally, it processes the fused results sequentially with shuffle block, depthwise separable convolutions, and ECA to create feature maps for heatmap generation and keypoint localization.

## 4. Experiments

In this section, we conduct experiments on two public benchmark datasets to evaluate MPE-HRNet.L. Firstly, we represent the details of datasets, training, and evaluation metrics in Section 4.1. Secondly, we analyze the ablation experiments in Section 4.2 and comparative experiments in Section 4.3, respectively. Finally, we discuss the limitation of HRNet.L in Section 4.4.

### 4.1. Experimental Settings

**Dataset.** We use two datasets, i.e., AP-10K [39] and Animal Pose [40], to evaluate our solution. **AP-10K dataset** is the a large-scale dataset for animal pose estimation. It includes 10,015 images collected from 23 families and 54 species of animals. The AP-10K dataset contains a total of 13,028 instances, each of which is labeled with 17 key points to describe poses. Table 2 shows the details of those keypoints. In our experiments, the AP-10K dataset is divided into train set, validation set, and test set. The three datasets include 7010 images, 1002 images, and 2003 images, respectively. **The Animal Pose dataset** contains 4608 images collected from dog, cat, cow, horse, and sheep species. It includes 6117 instances annotated with 20 keypoints. The Animal Pose dataset is also divided into train set, validation set, and test set. Each of these sets include 2798 images, 810 images, and 1000 images, respectively.

**Training**. All the training is carried out on the same server which is equipped with 5 NVIDIA GeForce RTX 3090 GPUs sourced from Santa Clara, CA, USA, and 376 GB of Kingston memory sourced from Fountain Valley, CA, USA. Our solution and all the comparative solutions are implemented in Python 3.8.19, and executed on PyTorch 2.5.0 framework. Multiple data augmentations are adopted in the training, which include random horizontal flip, random rotation from −80 to 80 degrees, random scaling by a factor of 0.5–1.5, and random body part augmentation generating animals with only the upper or lower part of a body.

To ensure the fairness of training processes, the training parameters, batch_size and the number of epochs, are separately set to 64 and 260, while all the other training parameters are configured with their default values. All parameters are updated by Adam optimizer combined with warm-up learning strategy. Learning rate is configured to $5\times 10^{-4}$ at the first 170 epochs and reduced to $5\times 10^{-5}$ and $5\times 10^{-6}$ at the 170th and 200th epochs, respectively.

**Evaluation metrics**. Object keypoint similarity (OKS) is a widely accepted criterion for pose estimation. It is calculated according to Equation (Equation 12). In the equation, $d_{i}$ describes the Euclidean distance between the model prediction and the ground truth of keypoint i, $v_{i}$ denotes the visibility flag of the ground truth of the key point, s represents the scale of the animal object, and $k_{i}$ describes the constant to control the fall-off of the keypoint.
(12)$$OKS=\frac{\sum_{i}e^{\frac{-d_{i}^{2}}{2s^{2}\hbar_{i}^{2}}}\delta \left(\nu_{i}>0\right)}{\sum_{i}\delta \left(\nu_{i}>0\right)}$$

We use the average precision (AP), $AP^{50}$ (AP at OKS = 0.5), $AP^{75}$ (AP at OKS = 0.75), $AP^{M}$ (AP for medium objects: 322 < area < 962), $AP^{L}$ (AP for large objects: area > 962), and average recall (AR) to evaluate the accuracy of models. AP is regarded as the main metric for accuracy. In addition, we use Params (M) to evaluate complexity, and use FLOPs (G) and inference time (ms) to evaluate speed.

### 4.2. Ablation Experiments

To analyze the effectiveness of the improvements in this work, we carry out a series of ablation experiments.

#### 4.2.1. Selection of SPPF^+^ and SPPF

SPPF^+^ fuses the feature maps generated by different pooling operations by add operation. Add operation adds feature maps related to the same channel. It can merge multiple groups of feature maps generated by different pooling operations into one group, thus reducing parameters for subsequent operations. However, it also results in feature loss. Fortunately, the feature loss could be alleviated if the feature maps involving the same add operation are much similar to each other.

For the feature maps generated by the stacked pooling operations in SPPF.+, the similarity between the feature maps related to the same channel increases with the increase in their resolutions. Therefore, SPPF.+ is more suitable for the branches of high resolutions. SPPF does not suffer from the issue. MPE-HRNet.L contains multiple branches of different resolutions. In order to improve accuracy through the proper use of SPPF.+ and SPPF in these branches, we carry out the experiments in which SPPF/SPPF.+ are added at the beginning of different branches.

Table 3 records the experiment results of Lite-HRNet and its variants only involving SPPF, SPPF.+, or both of them. +SPPF.1/8,1/16,1/32 denotes the variant with SPPF in the 1/8, 1/16, and 1/32 branches. +SPPF.+1/8 and SPPF.1/16,1/32 describes the variant version with SPPF.+ in the 1/8 branch and SPPF in the 1/16 and 1/32 branches. +SPPF.+1/8,1/16 and SPPF.1/32 represent the variant version with +SPPF.+ in the 1/8 and 1/16 branches and SPPF in the 1/32 branch. +SPPF.+1/8,1/16,1/32 indicates the variant version with SPPF.+ in the 1/8, 1/16, and 1/32 branches. From the experimental results, it can be observed that inserting SPPF.+ in the 1/8 and 1/16 branches and SPPF in the 1/32 branch yields the best results. In the following experiments, we configure SPPF and SPPF.+ according to the best results.

#### 4.2.2. Effectiveness Study of All the Improvements

We design five different models to evaluate the effectiveness of each improvement and describe the corresponding experiment results in Table 4. All these results are generated on the AP-10K dataset. In the table, Model0 denotes the original version of Lite-HRNet, while Model1 to Model4 describe different variants of the original version. Model1 and Model2 separately denote the variant with MFE-L in all the branches and the variant with MFE-L in 1/16 and 1/32 branches and MFE-H in 1/4 and 1/8 branches. Model3 represents the variant with MFE-L in 1/16 and 1/32 branches, MFE-H in 1/4 and 1/8 branches, and FE stage.

According to Table 4, Model1 and Model2 achieved improvements in AP, AR, parameters, and FLOPs(G) separately compared to Model0 and Model1, indicating that the introduction of MFE-L and MFE-H helped to improve the performance of Lite-HRNet. Model3 also achieved performance improvements in the above four evaluation metrics compared with Model2, which verified the effectiveness of the collaboration of MFE-L, MFE-H, and FE stage in improving the performance of Lite-HRNet.

Model4 describes the variant with MFE-L, MFE-H, the new stage, and $SPPF^{+}$/SPPF. It executes with a higher performance compared with Model3, which verified the effectiveness of all the improvements proposed in this work.

### 4.3. Performance Comparison

#### 4.3.1. Comparison on AP-10K Dataset

We carry out experiments to evaluate the detection accuracy and speed of MPE-HRNet.L by comparing it with nine important pose estimation solutions, including CSPNeXt-t [41], CSPNeXt-s, ShuffleNetV1 [20], ShuffleNetV2 [42], MobileNetV2 [43], MobileNetV3 [44], Dite-HRNet [45], Lite-HRNet, and DARK [46]. DARK utilizes Lite-HRNet to generate heatmaps but performs post-processing in a different way to the other solutions. All the solutions are lightweight solutions.

Table 5 shows the results of these solutions on the AP-10K validation set. According to the table, MPE-HRNet.L obtained the highest AP, $AP^{50}$, $AP^{75}$, $AP^{M}$, $AP^{L}$, and AR. Compared with ShuffleNetV2, ShuffleNetV1, MobileNetV2, MobileNetV3, CSPNeXt-t, Dite-HRNet, Lite-HRNet, DARK, and CSPNeXt-s, it improved AP in animal pose estimation by 8.8%, 8.5%, 6.4%, 5.5%, 5.3%, 2.3%, 2.2%, 2.0%, and 1.5%, respectively. It also achieved improvements when any of the other metrics was used. It is obvious that MPE-HRNet.L outperformed all the comparative solutions in accuracy.

We aimed to strike a balance between accuracy and complexity, striving to carefully manage the complexity of MPE-HRNet.L during its design process. In all the solutions, DARK and Lite-HRNet have the least Params, and Dite-HRNet has the lowest FLOPs. Comparing with these two solutions, MPE-HRNet.L does not increase Params and FLOPs too much while obtaining an improvement in accuracy. Concretely, it has only 0.43 M more parameters than DARK and Lite-HRNet, and 0.29 G more FLOPs than Dite-HRNet, respectively. Comparing with ShuffleNet, MobileNet, and CSPNeXt, MPE-HRNet.L has the least Params and the lowest FLOPs(G). The inference time of MPE-HRNet.L is 1.74 ms longer than that of DARK and 1.62 ms longer than Lite-HRNet, indicating a slight increase with negligible impact.

Table 6 shows the results on the AP-10K test set. According to the table, MPE-HRNet.L still achieved the highest AP in all the comparison solutions. Comparing with ShuffleNetV2, ShuffleNetV1, MobileNetV2, MobileNetV3, CSPNeXt-t, Dite-HRNet, Lite-HRNet, DARK, and CSPNeXt-s, it improved AP by 8.4%, 7.9%, 4.9%, 4.9%, 4.6%, 1.4%, 1.3%, 1.4%, and 1.1%, respectively. MPE-HRNet.L also exhibited the highest $AP^{50}$, $AP^{75}$, $AP^{M}$, $AP^{L}$, and AR in all the other comparison solutions except DARK. Although DARK surpasses our method in the $AP^{M}$ metric, it falls behind MPE-HRNet.L in all the other metrics. Considering all the metrics, MPE-HRNet.L outperforms all comparison solutions on the test set.

We also compared MPE-HRNet.L with ResNet-101, HRNet-W32, and RFB-HRNet. The latter three solutions are based on large models. According to the results in Table 7, these three solutions exhibit a higher accuracy than MPE-HRNet.L. However, they also have many more Params than MPE-HRNet.L. In many practical application scenarios, pose estimation models are required to be deployed on edge devices. Large models struggle to perform optimally on edge devices due to the inability of these devices to provide the resources required by the models.

#### 4.3.2. Comparison on Animal Pose Dataset

We also conducted experiments on the Animal Pose dataset. In these experiments, we compared MPE-HRNet.L with ShuffleNetV2, MobileNetV2, and CSPNeXt-s. The corresponding results are recorded in Table 8. Similar to the results on the AP-10K dataset, MPE-HRNet.L also achieved the highest AP on the Animal Pose dataset, demonstrating the effectiveness of MPE-HRNet.L in improving accuracy.

### 4.4. Limitation Discussion

While MPE-HRNet.L obtains accuracy improvements, it still suffers from two obvious limitations. The first limitation is that MPE-HRNet.L does not perform well in complex occlusion scenes. The second limitation is that the accuracy of MPE-HRNet.L needs to be improved.

**Complex occlusion limitation.** We chose twelve images with different scenes and depicted the corresponding results of MPE-HRNet.L in Figure 7. These images were shot at different distances and angles. They include various animals belonging to different species. Some of them even include shadow and object occlusions. According to Figure 7b,e,h, MPE-HRNet.L performed well in the images including multiple objects even if those objects are different sizes. It also performed well in the scenario where shadow exists, according to Figure 7i. MPE-HRNet.L can still work well in the scenario where some parts of bodies are occluded lightly, as Figure 7j,k show. However, it does not work well in the scenario where complex occlusion exists, as Figure 7f shows.

**Accuracy limitation.** According to Section 4.3.1 and Section 4.3.2, MPE-HRNet.L obtains accuracy improvements compared with lightweight models such as Dark, Lite-HRNet, and so on. However, MPE-HRNet.L exhibits a relatively low accuracy when compared with the solutions based on large models. In addition, MPE-HRNet.L improves accuracy by 1.4%, 1.3%, and 1.4% compared with Dark, Lite-HRNet, and Dite-HRNet, respectively. However, it increases parameters by 38% (0.43 M), 38% (0.43 M), and 31% (0.37 M) compared with the three solutions, respectively. Compared to the improvement in accuracy, the increase in parameters is more pronounced. Nonetheless, the number of parameters of MPE-HRNet.L is only 1.56 M. MPE-HRNet.L is still a lightweight model. However, MPE-HRNet.L does not exhibit any advantage in accuracy compared to large models. It is necessary to improve MPE-HRNet.L to obtain higher accuracy.

## 5. Conclusions

In this work, we focused on the problem of estimating the poses of multiple species with high accuracy in real scenarios. To deal with this problem, we constructed a lightweight network by improving Lite-HRNet. Concretely, we proposed SPPF.+ and applied it as well as SPPF to all branches. We proposed MPM and DSCA, and constructed MFE block by applying them together. We also designed a new stage to enhance the features in the feature map output by MPE-HRNet.L. The experimental results on the AP-10K dataset and the Animal Pose dataset verified the effectiveness of our work. In the future, we will focus on the problem of animal pose estimation involving complex occlusions.

## Figures and Tables

**Figure 1 sensors-24-06882-f001:**
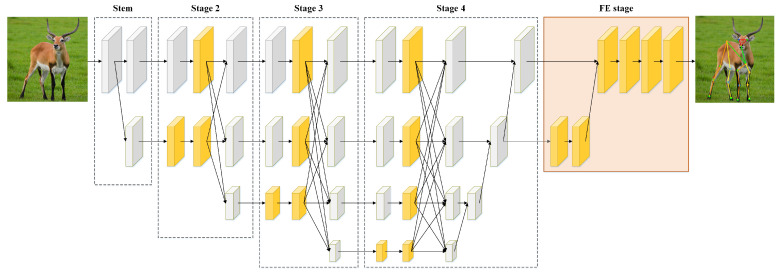
Overall architecture of MPE-HRNet.L. MPE-HRNet.L includes five stages. MFE block and SPPF.+/SPPF are applied to stages 2 to 4. FE stage is appended as the final stage. The yellow cuboids describe the feature maps processed by our improvements.

**Figure 2 sensors-24-06882-f002:**
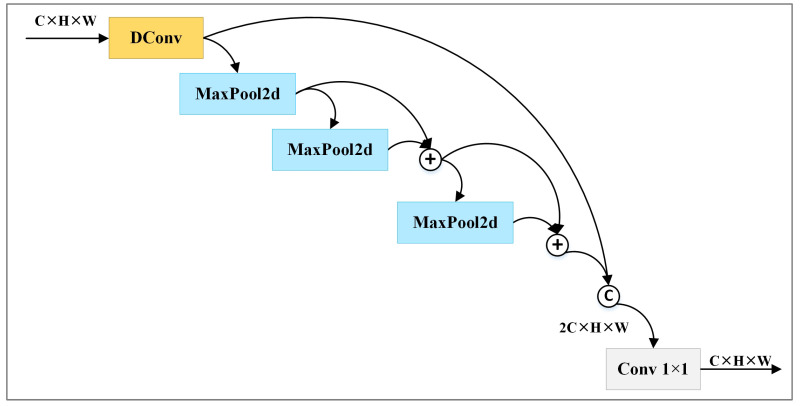
Structures of SPPF.+. © and ⊕ represent concat operation and add operation, respectively.

**Figure 3 sensors-24-06882-f003:**
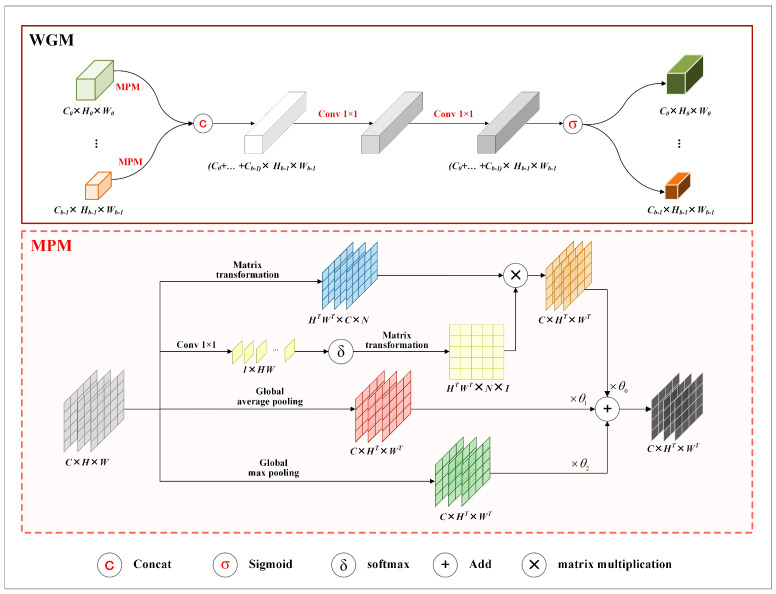
Structure of WGM. The top part shows the structure of WGM, and the bottom part describes the structure of MPM. Matrix transformation represents a series of operations, e.g., reshaping, unsqueezing, and permuting, on the dimensions or sizes of three−dimensional matrices. $H^{T}$ and $W^{T}$ denote the height and width of a transformed matrix, and N is calculated by dividing $H^{T}W^{T}$ by HW.

**Figure 4 sensors-24-06882-f004:**
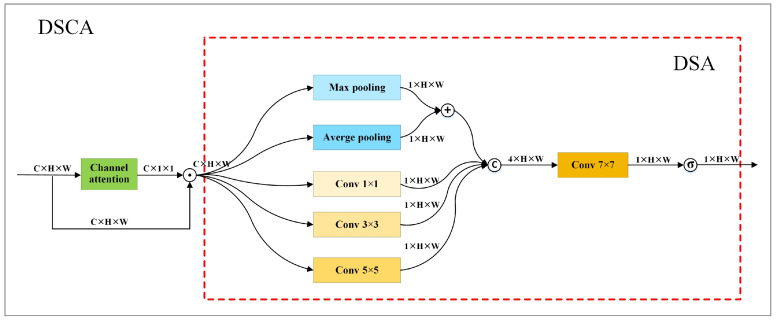
Structure of DSCA. It comprises a channel attention module and a DSA module. The area surrounded by red dashed lines describes the structure of DSA. ⨀ represents element-wise multiplication.

**Figure 5 sensors-24-06882-f005:**
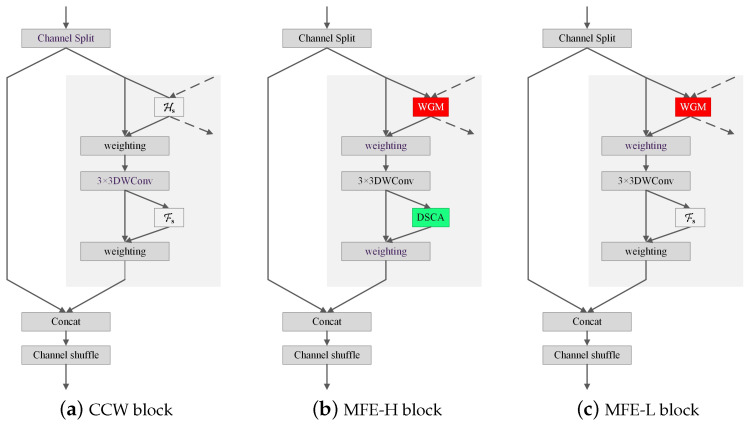
Structures of MFE. Both MFE-H and MFE-L exploit WGM to generate weighted feature maps as the input of the 3 × 3 DWConv. MFE-H also utilizes the proposed DSCA to generate the weighted feature maps involving in the concat operation, while MFE-L still utilizes $F_{s}$ to generated weighted feature maps.

**Figure 6 sensors-24-06882-f006:**
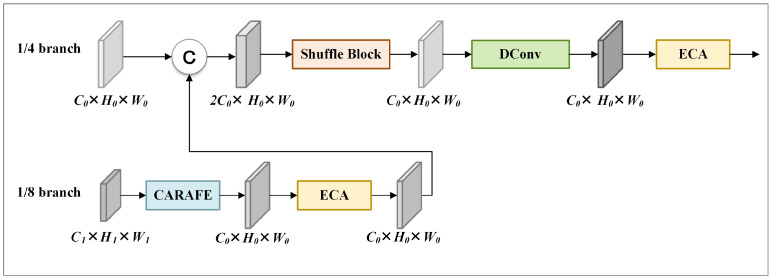
Structure of FE stage. The cuboids represent feature maps. © denotes concat operation.

**Figure 7 sensors-24-06882-f007:**
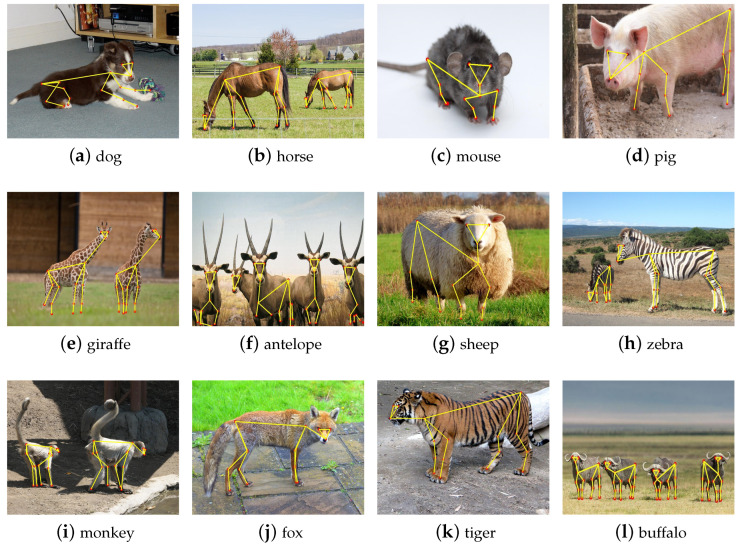
Example qualitative results of the AP-10K.

**Table 1 sensors-24-06882-t001:** Structure details of MPE-HRNet.L.

Stages	Output Size	Operations	Resolution Branch	Output_Channels	Repeat	Num_Modules
stem	64 × 64	conv2d	2 ×	32	1	1
		shuffle block	4 ×	32	1	1
2	64 × 64	SPPF.+	4 × 8 ×	40, 80	1	1
		MFE block	4 × 8 ×	40, 80	2	2
		fusion block	4 × 8 ×	40, 80	1	
3	64 × 64	SPPF.+	4 × 8 × 16 ×	40, 80, 160	1	1
		MFE block	4 × 8 × 16 ×	40, 80, 160	2	2
		fusion block	4 × 8 × 16 ×	40, 80, 160	1	
4	64 × 64	SPPF	4 × 8 × 16 × 32 ×	40, 80, 160	1	1
		MFE block	4 × 8 × 16 × 32 ×	40, 80, 160	2	2
		fusion block	4 × 8 × 16 × 32 ×	40, 80, 160	1	
FE	64 × 64	upsampling	4 ×	80	1	1
		shuffle block	4 ×	80	1	
		DConv	4 ×	40	1	
		ECA	4 ×	40	1	

**Table 2 sensors-24-06882-t002:** Definitions of keypoints.

Keypoint	Definition	Keypoint	Definition
0	Left Eye	9	Right Elbow
1	Right Eye	10	Right Front Paw
2	Nose	11	Left Hip
3	Neck	12	Left Knee
4	Root of Tail	13	Left Back Paw
5	Left Shoulder	14	Right Hip
6	Left Elbow	15	Right Knee
7	Left Front Paw	16	Right Back Paw
8	Right Shoulder		

**Table 3 sensors-24-06882-t003:** Effectiveness study of SPPF.+ and SPPF.

Models	AP	AR	Params (M)	FLOPs (G)
Lite-HRNet	59.8	64.9	1.13	0.35
+SPPF.18,116,132	61.2	65.9	1.60	0.63
+SPPF.+18 & SPPF.116,132	61.3	66.0	1.58	0.63
+SPPF.+18,116 & SPPF.132	62.0	66.3	1.56	0.62
+SPPF.+18,116,132	61.2	66.0	1.46	0.62

**Table 4 sensors-24-06882-t004:** Effectiveness study of all the improvements.

Models	MFE-L^*all*^	MFE-L.1/16,1/32 & MFE-H.1/4,1/8	FE Stage	SPPF^+^ & SPPF	AP	AR	Params (M)	FLOPs (G)
model0					59.8	64.9	1.13	0.35
model1	✓				60.1	65.0	1.13	0.36
model2		✓			60.4	65.2	1.17	0.44
model3		✓	✓		61.0	65.6	1.25	0.58
model4		✓	✓	✓	62.0	66.3	1.56	0.62

SPPF.+ and SPPF means inserting SPPF.+ and SPPF to the 1/8 and 1/16 branches, and the 1/32 branch, respectively.

**Table 5 sensors-24-06882-t005:** Results on the AP-10K validation set.

Methods	AP	AP^50^	AP^75^	AP^*M*^	AP^*L*^	AR	Inference Time (ms)	Params (M)	FLOPs (G)
CSPNeXt-t	56.7	87.1	58.2	39.2	57.0	61.5	9.74	6.02	1.91
CSPNeXt-s	60.5	88.7	63.2	47.3	60.7	64.8	11.23	8.58	2.36
DARK	60.0	89.1	63.2	46.2	60.3	64.7	27.27	1.13	0.35
Dite-HRNet	59.7	89.1	61.9	48.0	60.0	64.8	33.61	1.19	0.33
Lite-HRNet	59.8	88.9	63.2	51.5	60.1	64.9	27.39	1.13	0.35
ShuffleNetV1	53.5	84.9	54.0	40.9	53.8	58.9	13.61	6.94	1.80
ShuffleNetV2	53.2	83.9	53.0	36.3	53.6	58.7	12.10	7.55	1.83
MobileNetV2	55.6	86.8	57.5	41.7	55.9	61.2	12.27	9.57	2.12
MobileNetV3	56.5	87.3	58.9	48.8	56.8	61.3	13.93	5.24	1.73
MPE-HRNet.L	62.0	90.1	65.6	53.9	62.3	66.3	29.01	1.56	0.62

**Table 6 sensors-24-06882-t006:** Results on the AP-10K test set.

Methods	AP	AP.50	AP.75	AP.M	AP.L	AR	Inference Time (ms)	Params (M)	FLOPs (G)
CSPNeXt-t	55.4	86.2	58.4	46.0	55.9	60.9	10.29	6.02	1.91
CSPNeXt-s	58.9	87.7	62.5	49.2	59.5	63.8	11.84	8.58	2.36
DARK	58.6	87.2	61.0	52.1	59.1	64.0	30.02	1.13	0.35
Dite-HRNet	58.6	87.2	62.1	51.3	59.1	64.2	36.17	1.19	0.33
Lite-HRNet	58.7	87.4	61.9	48.3	59.4	64.0	28.14	1.13	0.35
ShuffleNetV1	52.1	83.6	52.5	42.0	52.7	57.9	13.24	6.94	1.80
ShuffleNetV2	51.6	83.4	51.0	45.0	52.2	57.6	12.21	7.55	1.83
MobileNetV2	55.1	85.8	56.1	41.6	55.7	60.7	12.17	9.57	2.12
MobileNetV3	55.1	86.1	57.3	47.6	55.6	60.5	13.56	5.24	1.73
MPE-HRNet.L	60.0	88.7	62.9	51.4	60.5	65.0	28.03	1.56	0.62

**Table 7 sensors-24-06882-t007:** Comparison with solutions based on large models.

Methods	AP	AP.50	AR	Params (M)	FLOPs (G)
ResNet-101	68.1	93.8	77.4	52.99	12.13
HRNet-W32	72.2	94.2	77.0	28.54	10.25
RFB-HRNet	75.0	95.8	78.1	63.97	22.61
Ours	62.0	90.1	66.3	1.56	0.62

The results of RFB-HRNet come from the work of Gong et al. [19].

**Table 8 sensors-24-06882-t008:** Results on the Animal Pose dataset.

Methods	AP	AP.50	AP.75	AP.M	AP.L	AR	Inference Time (ms)	Params (M)	FLOPs (G)
CSPNeXt-s	61.1	90.3	68.4	61.2	61.3	65.9	8.67	8.58	2.36
DARK	63.3	90.5	70.7	65.0	63.2	68.5	27.47	1.13	0.35
Dite-HRNet	62.0	90.1	67.0	63.3	62.0	67.3	30.10	1.19	0.33
Lite-HRNet	62.2	89.4	68.3	64.4	62.1	67.5	26.47	1.13	0.35
ShuffleNetV2	56.5	86.4	61.8	59.5	56.0	61.9	12.55	7.55	1.83
MobileNetV2	58.7	88.1	64.6	63.1	58.3	64.0	12.07	9.57	2.12
MPE-HRNet.L	64.9	90.5	72.3	65.9	64.9	69.8	28.17	1.56	0.62

## Data Availability

The datasets used in this article are publicly available. The AP-10K dataset can be accessed at https://drive.google.com/file/d/1-FNNGcdtAQRehYYkGY1y4wzFNg4iWNad/view (accessed on 15 May 2024), and the Animal Pose dataset is available at https://openxlab.org.cn/datasets/OpenDataLab/animalpose (accessed on 4 October 2024).

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
