# Peer review of "MPE-HRNetL: A Lightweight High-Resolution Network for Multispecies Animal Pose Estimation"

_sensors, 2024, doi:10.3390/s24216882_

Round 1
Reviewer 1 Report
Comments and Suggestions for Authors
The work focuses on recognizing the position of key points on the body of animals, and hence animal poses, using computer vision. A fairly large publicly available dataset AP-10K is used. The authors perform a slight tuning of the existing neural network architecture to create a model that is both accurate and lightweight. Unfortunately, the work contains significant shortcomings and can only be considered for publication after very careful revision and repeated review.
Major comment:
The authors claim that the advantage of their model and their architecture is that it is both lightweight and accurate, and it has an extremely small number of parameters. At the same time (the authors do not specify this) the accuracy achieved by the authors is very low, compared to existing works using the same dataset and solving the same problem. It needs to be explained in more detail (the very short explanation given is unsatisfactory) why we should create such lightweight models and limit ourself. This leads to the fact that the accuracy achieved is very low. For example, in the paper https://www.mdpi.com/2079-9292/12/20/4210 the AP value (in the authors' paper 60-62) is 77 and other metrics are also higher. In the paper https://www.mdpi.com/2079-9292/11/22/3702 the HRNet architecture and modifications are used for the same problem, using the same dataset. Yes, the cited paper has 23-64M parameters instead of 1.6M, but AP is also 69-75. The authors point out “it cannot be deployed on the devices with limited computing power, whereas these devices are suitable for real pose estimation environments.” - is it possible to give any numbers and any justification? How many maximum parameters can be used? Yes, the authors' model has 20-50 times fewer parameters, but in what situations does it really matter? In addition, it is important to explicitly state that the achieved accuracy is low ("far from the "state of the art"), but at the expense of this and the parameters are few.
Table 6 shows that Lite-HRNet/Dite-HRNet have a slightly worse accuracy than the authors, but they have fewer parameters. Is it the authors' goal to minimize the number of parameters, then exactly is this a good comparison? Some discussion on this issue needs to be added. The authors write “MPE-HRNet L did not increase Params and FLOPs too much while obtaining obvious improvements in accuracy”. The improvement in accuracy from 59 to 60 (AP) is not too much, and the number of parameters are 38% larger. It seems that this phrase is incorrect. Above when I write "parameters" it can be replaced with "FLOPs" - the questions will be the same.
Authors need to add a comparison of accuracy and number of parameters with previous models solving the same problem and using the same dataset. Besides it is necessary to add a discussion about why it is so important to use it on low-power equipment: in what situation? What is the frame rate? Why exactly? General and vague wording should be avoided.
Minor comments:
1) The authors describe the architectures (developed earlier) in too much detail, in many places it is enough to put a reference. Some of the figures should be removed or condensed and merged.
2) Capitalize the titles of the sections “architecture of MPE-HRNet L”, “attention mechanism”.
3) The sections “Related Works” and “Introduction” duplicate each other, and some phrases are repeated almost exactly: “Wang et al. [17] designed a grouped pig pose estimation model named as GANPose” and ”Wang et al. [17] used HRNet to estimate the poses of pigs”. Duplication should be eliminated, also the reviewer has doubts about the need for a “Related Works” section.
4) More articles citing and using the AP-10K dataset should be cited, in particular articles that performed similar classification. Also perhaps the authors should look at the impressive work of https://doi.org/10.21203/rs.3.rs-4687765/v1 using the same dataset but extremely small number of labeled images for training.
The manuscript may be reconsidered after major revision, adding an explanation of the novelty and importance of using lightweight models specifically for this task, and significantly reducing the text.
Reviewer 2 Report
Comments and Suggestions for Authors
This paper constructs a lightweight neural network model for multi-species animal pose estimation, improve Lite-HRNet7, and propose MPE-HRNetL. Experiments show that the proposed method has good performance on AP-10K data set. However, the writing of this article needs to be strengthened. In order to improve the readability of the paper, there are some opinions and suggestions worth sharing and areas that need to be revised.
1. In the first paragraph of Introduction, bottom-up methods and top-down methods are need to be further introduced for readers' convenience.
2. Brief introduction at the beginning of each section: Ensure that each section starts with a brief overview of its content. For example,“This section will discuss the implementation process of the experiments. Firstly.....”
3. Incorrect punctuation on page 2, line 59.
4. Explanation the full name of AP on Page 2, Line 65.
5. Repetitions and repeated references in Section 2.1 and Introduction, such as the description of the reference [9]. Identify and remove duplicate references and descriptions. Ensure that the Introduction provides a general overview, while Section 2.1 delves into detailed related works.
6. Abbreviation for SPP-Fast on Page 3, Line 137. If SPP-Fast and Spatial Pyramid Pooling Fast refer to the same network, use one consistent abbreviation. If they are different, clearly distinguish between them.
7. Explanation the full name of FE abbreviation on Page 4, Line 146.
8. Enlarge Figure 6 for better visibility and provide a detailed caption explaining the key elements illustrated in the figure.
9. Specify the deep learning framework (e.g., TensorFlow, PyTorch) and the programming language (e.g., Python) used in your training process.
10. Add a description of the limitations of the proposed method in the experimental section.
Comments on the Quality of English LanguageEnglish language quality needs to be improved, especially in the introduction and related works sections.
Reviewer 3 Report
Comments and Suggestions for Authors
The manuscript “MPE-HRNet^L : A Lightweight High-Resolution Network for Multi-Species Animal Pose Estimation” mainly studies animal pose estimation, and propose an improved Lite-HRNet which named as MPE-HRNet^L. An improved version of SPPF is proposed to mitigate the impact of feature loss caused by pooling operations. A mixed pooling module and a dual spatial attention mechanism are proposed to design a mixed feature extraction (MFE) block with a stronger feature extraction capability. The proposed method appears to offer improved accuracy over existing lightweight models while maintaining low computational requirements. The manuscript is generally well-structured and clearly written. The figures are well-presented and aid in understanding the proposed system.
Author Response
Thank you for your positive feedback on our manuscript "MPE-HRNetL: A Lightweight High-Resolution Network for Multi-Species Animal Pose Estimation." We appreciate your recognition of our work's structure, clarity, and effectiveness.
Your encouraging comments motivate us to continue contributing meaningfully to the field. If you have any further suggestions or requests for improvements, please feel free to share.
Thank you again for your time and consideration.
Reviewer 4 Report
Comments and Suggestions for Authors The authors proposed a Lightweight High - Resolution Network for Multi - Species Animal Pose Estimation. The entire research is relatively novel, and the writing structure is reasonable, possessing certain scientific value. However, prior to acceptance, there are still some issues that need to be addressed.Comments on the Quality of English Language
Moderate editing of English language required.
Round 2
Reviewer 1 Report
Comments and Suggestions for Authors
I think the authors for answering my questions. Now the manuscript can be accepted in its current form.